# Anisotropic Diffusion Based Multiplicative Speckle Noise Removal

**DOI:** 10.3390/s19143164

**Published:** 2019-07-18

**Authors:** Mei Gao, Baosheng Kang, Xiangchu Feng, Wei Zhang, Wenjuan Zhang

**Affiliations:** 1School of Information Science and Technology, Northwest University, Xi’an 710127, China; 2School of Science, Xi’an Technological University, Xi’an 710021, China; 3Department of Mathematics, Xidian University, Xi’an 710071, China

**Keywords:** anisotropic diffusion, partial differential equations (PDE), multiplicative noise removal

## Abstract

Multiplicative speckle noise removal is a challenging task in image processing. Motivated by the performance of anisotropic diffusion in additive noise removal and the structure of the standard deviation of a compressed speckle noisy image, we address this problem with anisotropic diffusion theories. Firstly, an anisotropic diffusion model based on image statistics, including information on the gradient of the image, gray levels, and noise standard deviation of the image, is proposed. Although the proposed model can effectively remove multiplicative speckle noise, it does not consider the noise at the edge during the denoising process. Hence, we decompose the divergence term in order to make the diffusion at the edge occur along the boundaries rather than perpendicular to the boundaries, and improve the model to meet our requirements. Secondly, the iteration stopping criteria based on kurtosis and correlation in view of the lack of ground truth in real image experiments, is proposed. The optimal values of the parameters in the model are obtained by learning. To improve the denoising effect, post-processing is performed. Finally, the simulation results show that the proposed model can effectively remove the speckle noise and retain minute details of the images for the real ultrasound and RGB color images.

## 1. Introduction

Image denoising is a very important problem in image processing. A variety of image denoising methods have been developed to deal with additive noise in recent decades, including variational-based methods [1,2], partial differential equation (PDE)-based methods [3,4,5,6], filter-based methods [7,8], sparse representation (SR)-based methods [9,10,11], wavelet-based methods [12,13], deep neural network (DNN)-based methods [14,15,16], etc. However, multiplicative noise is still a difficult problem to deal with it by various methods. Due to the special structure of multiplicative noise, we hope to gain more knowledge and insight. Therefore, methods including variational-based [17,18,19,20,21,22], PDE-based [23,24,25,26], filter-based [27,28,29], SR-based [30,31,32], wavelet-based [33,34,35] and DNN-based [36,37,38], low rank approximation-based [39,40] are all studied in parallel. In particular, although the effect of method proposed by Zhou et al. based on PDE in 2015 and 2017 [41,42] is very good, there is still some room for improvement. Motivated by Zhou’s work, we focus on anisotropic diffusion-based multiplicative speckle noise removal methods in this work.

In past decades, multiplicative noise removal has attracted much attention. Multiplicative noise, also known as speckle noise, widely exists in the real world image such as the nuclear magnetic resonance images [43], remote sensing images [9], synthetic aperture radar images [23,24,44,45,46], medical ultrasonic images [27,28,29,35,47,48], and so on. It is well known that multiplicative speckle noise can reduce the quality of images, seriously affect image segmentation, classification, target detection and extraction of other regions of interest. Compared with additive noise, speckle noise has a more serious effect on structure and performance. Therefore, research into multiplicative speckle noise removal is a very important issue in image processing. The principle of speckle noise can be described as follows. Let I0 denote the observed noisy image defined on a rectangle Ω⊂ℝ2, I be the corresponding true image and n be the noise term that is independent of I. Then, speckle noise images can be modeled as follows:(1)I0=In.

Speckle noise is difficult to remove from noisy images due to its multiplicative properties. However, images from ultrasound devices usually possess the property of logarithmic compression [18,47,49]. Experimental measurements show that the compressed speckle noise model can be described as follows [47]:(2)$$I_{0}=I+\sqrt{I}n,$$
where n is a Gaussian noise with a mean of zero and standard deviation σ. Thus, speckle noise can be regarded as a form of additive noise whose noise is proportional to the square root of grayscale.

At present, deep learning has almost become the standard in the field of computer vision and the hottest research direction in the field of artificial intelligence. However, deep learning is completely driven by data and we now know the noise imaging mechanism of the model, we can add noise to the original image in the synthetic image experiment, and then denoising. If this knowledge is gained by machine learning, it will be laborious. At the same time, if the noise distribution is not like the Gaussian distribution in post-processing, we can adjust it to Gaussian, and substitute it back into the imaging noise model to solve images that cannot be learned by machine learning.

PDE-based nonlinear diffusion methods have been widely used in additive noise and multiplicative noise removal. In particular, anisotropic diffusion methods have received extensive attention due to their advantages of preserving and even enhancing the edge information in image denoising. Anisotropic diffusion was first proposed by Perona and Malik (1990) [50], which is also known as the PM model. The PM model removes the noise with smaller gradient changes, while it preserves edges with larger gradient changes by using a gradient operator to identify noise and edge caused by the change of image gradient. Since the work of Perona and Malik (1990), various anisotropic diffusion methods have been proposed in the field of additive noise removal. For example, Yu and Acton (2002) proposed the speckle reducing anisotropic diffusion (SRAD) method [51], which can handle various noise distributions. As a further improvement of the SRAD method, Karl et al. (2007) proposed the oriented speckle reducing anisotropic diffusion (OSRAD) method [6], introducing local directional variance of image intensity. SRAD and OSRAD remove speckle from the entire image, thus resulting in piece-wise smooth images, called over-smooth images. To solve this over-smooth problem, [29] gives a solution; that is, anisotropic diffusion with memory based on speckle statistic (ADMSS) method [29]. Zhou et al. [41] proposed a doubly degenerate nonlinear diffusion (DDND) model [41] by using the diffusion equation theory to guide the denoising process with the aid of the gradient information and the gray level information. Through further research, Zhou et al. [42] proposed a nonlinear diffusion equation-based (NDEB) model, using the region indicator as the variable exponent. With regard to anisotropic diffusion denoising, we first proposed a new class of fractional-order anisotropic diffusion equations for image denoising in 2007 [52]. On the basis of this work, a class of generalized anisotropic diffusion equations for image denoising is proposed in 2017 [53]. 

The goal of this paper is to develop anisotropic diffusion models for solving the multiplicative speckle noise removal problem. The main work we have done in this paper is as follows. Firstly, based on the characteristic of speckle noise and inspired by the DDND model [41], an indicator function of gray level and noise standard deviation is introduced as diffusion coefficient in the PM model, which is used to steer the whole denoising process. The proposed model can effectively remove speckle noise and retain small details, especially in the low-contrast and low gray level and low noise standard deviation regions. However, it does not take into account the noise at the edge of the image. Therefore, we proposed an improved model. Secondly, because there is no ground truth in the real image experiment, two new iteration-stopping criteria are given: kurtosis is used to measure whether the noise is close to Gaussian noise, and correlation is used to measure the independence of the experimental image and noise, so as to stop iteration. The parameters in the model are trained. Considering their independence from noise variance, we use regression to find out the optimal parameter value through learning. In addition, the deviation is corrected. If kurtosis is used as the stopping criterion in the experiment, when the kurtosis of the noise is zero, its histogram is not necessarily Gaussian. Therefore, we do histogram specification of the noise so that the adjusted noise is closer to the required Gaussian distribution. Finally, we perform post-processing to obtain a denoising image by bringing the adjusted noise back to the model.

The rest of this paper is organized as follows. In Section 2, we briefly introduce the PM model and the DDND model, and give the motivation for our models. In Section 3, we propose an anisotropic diffusion model and its improved model for multiplicative speckle noise removal. Our experimental scheme is described in Section 4. The experimental results and discussions of synthetic images, real ultrasound images and RGB color images are included in Section 5. Finally, a brief conclusion is presented in Section 6.

## 2. Motivation

### 2.1. Classical Perona-Malik Model

Perona and Malik (1990) [50] proposed the following anisotropic diffusion equation: (3)$$\{\begin{array}{l} \frac{\partial I}{\partial t}=div\left[c\left({\left|\nabla I\right|}^{2}\right)\cdot \nabla I\right]\;\;on\;\;\Omega \times \left(0,T\right)\\ \frac{\partial I}{\partial N}=0\;\;on\;\partial \Omega \times \left(0,T\right)\\ I\left(x,0\right)=I_{0}\left(x\right)\;on\;\Omega \end{array},$$
where ∇ and div stand for the gradient operator and the divergence operator, respectively, |·| denotes the magnitude, N denotes the outer normal vector, $I_{0}$ is the initial image, and c(·) denotes the diffusion coefficient. 

Although the PM model can eliminate noise and retain features, it has the following two drawbacks. First, when the image is contaminated by strong noise, the gradient change caused by noise is likely to exceed the gradient change caused by the edge. Thus, the result of diffusion cannot effectively remove the noise, but may enhance the noise. Second, the image after denoising usually has obvious “ladder” effect. As shown in Figure 1, the PM model is employed to remove speckle noise in a Barbra image (256 × 256 pixels) (Barbra image is available at (http://sipi.usc.edu/services/data-base/index.html)). It retains the sharp edges but generates some isolated black and white noise points in the restored image.

### 2.2. Doubly Degenerate Nonlinear Diffusion Model

According to Model (2), for arbitrary $x_{0}\in \Omega$, the expectation of $I_{0}\left(x_{0}\right)$ is
(4)$$E\left(I_{0}\left(x_{0}\right)\right)=E\left(I\left(x_{0}\right)+\sqrt{I\left(x_{0}\right)}n\left(x_{0}\right)\right)=I\left(x_{0}\right),$$
and the variance of $I_{0}\left(x_{0}\right)$ is
(5)$$V\left(I_{0}\left(x_{0}\right)\right)=E\left(I_{0}{\left(x_{0}\right)}^{2}\right)-E{\left(I_{0}\left(x_{0}\right)\right)}^{2}=I\left(x_{0}\right)\sigma^{2}.$$

Obviously, the variance of the compressed speckle noisy image depends on the gray level of the original image. In addition, multiplicative speckle noise has a more serious effect on structure and damage than additive noise. Consequently, different denoising strategies should be used according to the gray level of the image. The DDND model [41] takes the following form:(6)$$\{\begin{array}{l} \frac{\partial I}{\partial t}=\mathrm{div}\left[d\left(I\right)\frac{1}{{\left(1+{\left|\nabla I\right|}^{2}\right)}^{\left(1-\beta \right)/2}}\nabla I\right]\;\;on\;\;\Omega \times \left(0,T\right)\\ \frac{\partial I}{\partial N}=0\;\;on\;\partial \Omega \times \left(0,T\right)\\ I\left(x,0\right)=I_{0}\left(x\right)\;on\;\Omega \end{array}$$
where $d\left(I\right)=\frac{2{\left|I\right|}^{\alpha }}{M^{\alpha }+{\left|I\right|}^{\alpha }}$ is the gray level indicator function with α>0, $M=\sup_{x\in \Omega }I\left(x\right)$, and 0<β<1. 

The above model uses the gray level indicator function d(I) to extract the gray level information in images contaminated with speckle noise and steer the denoising process in different gray level regions, which has proven to be an effective denoising method.

## 3. Our Proposed Model

It can be seen from Equations (4) and (5) that the standard deviation of the compressed speckle noisy image not only depends on the square root of the grayscale of the original image, but also depends on the standard deviation of noise. Inspired by the above observation, we construct a new gray level indicator function:(7)$$b\left(I\right)=\frac{2{\left|\sqrt{I}\sigma \right|}^{\alpha }}{M^{\alpha }+{\left|\sqrt{I}\sigma \right|}^{\alpha }}.$$

By replacing d(I) in (6) with b(I), we obtain a new anisotropic diffusion equation:(8)$$\{\begin{array}{l} \frac{\partial I}{\partial t}=\mathrm{div}\left[b\left(I\right)\frac{1}{{\left(1+{\left|\nabla I\right|}^{2}\right)}^{\left(1-\beta \right)/2}}\nabla I\right]\;\;on\;\;\Omega \times \left(0,T\right)\\ \frac{\partial I}{\partial N}=0\;\;on\;\partial \Omega \times \left(0,T\right)\\ I\left(x,0\right)=I_{0}\left(x\right)\;on\;\Omega \end{array}$$
where α>0, 0<β<1 and $M=\sup_{x\in \Omega }\sqrt{I}\sigma$. It is worth noting that *M* here is the maximum value of noise standard deviation.

For one thing, we consider the influence of parameters α and β. The parameter α, which comes from the gamma correction [54], is used to release speckle noise compression information. Similar to the work of [41], the proposed function $b\left(I\right)=\frac{2{\left|\sqrt{I}\sigma \right|}^{\alpha }}{M^{\alpha }+{\left|\sqrt{I}\sigma \right|}^{\alpha }}$ can be transformed into $b\left(J\right)=\frac{2J^{\alpha }}{1+J^{\alpha }}$, where $J=\frac{\left|\sqrt{I}\sigma \right|}{M}\epsilon \left[0,\;1\right]$. As illustrated in Figure 2, different values of α will expand the values in different areas to release the compressed information. As for the parameter β in function $c\left({\left|\nabla I\right|}^{2}\right)=\frac{1}{{\left(1+{\left|\nabla I\right|}^{2}\right)}^{\left(1-\beta \right)/2}}$, according to the theory of diffusion equation, it plays the role of controlling the speed of degeneracy.

For another thing, we sketch the mechanism of diffusion process. The gray level indicator function $b\left(I\right)=\frac{2{\left|\sqrt{I}\sigma \right|}^{\alpha }}{M^{\alpha }+{\left|\sqrt{I}\sigma \right|}^{\alpha }}$ steers the whole diffusion progress. It can obviously be seen that b(I) becomes very small in the low noise standard deviation regions, which causes the diffusion coefficient to be close to 0 and maintains the low gray level image features. In the high grayscale regions, b(I) approaches 1, the diffusion coefficient is mainly controlled by the function $c\left({\left|\nabla I\right|}^{2}\right)=\frac{1}{{\left(1+{\left|\nabla I\right|}^{2}\right)}^{\left(1-\beta \right)/2}}$. It is easy to see that $c\left({\left|\nabla I\right|}^{2}\right)\to 0$ as |∇I|→∞, that is, near the boundaries of the region where the magnitude of gradient is large, the diffusion is “stopped” and the edges are preserved, and $c\left({\left|\nabla I\right|}^{2}\right)\to 1$ as |∇I|→0, that is, inside the regions where the magnitude of the gradient is weak, resulting in the isotropic diffusion. In this case, $c\left({\left|\nabla I\right|}^{2}\right)$ is an edge detector, which is used to preserve the edge of the image.

Although the new model is similar in form to the DDND model, they have essential differences. It is worth emphasizing that there are two advantages to the explicit representation of noise standard deviation in the model. Firstly, an extreme example is that when the noise standard deviation is 0, our model degenerates to 0 at the right end, i.e., $\frac{\partial I}{\partial t}=0$, I remains motionless, and we get a constant operator, while DDND does not have this ability. Secondly, the parameter of the noise standard deviation is robust. In other words, our parameter α is not dependent on parameter σ, parameter β in the PM model is not dependent on standard deviation σ. Thus, our parameters are robust to the noise standard deviation, while the parameters in the DDND model are dependent on the noise standard deviation, which is the most important difference between our work and DDND.

### Improved Model

As mentioned above, the diffusion is “stopped” near boundaries of the region, where |∇I| is large and the isotropic diffusion “takes place” inside the regions where |∇I| is small. However, it is not good that the diffusion is “stopped” near boundaries of the region, since there exists noise near the boundaries of the region. Thus, we hope that at the edges, diffusion occurs along the boundaries rather than along the vertical direction of the boundaries. For this purpose, we first decompose the divergence term of Equation (3) using the local image structure, and then combine it with the gray level indicator function constructed previously in the following work.

In fact, if we introduce two arbitrary orthonormal directions η and ξ, we have $\Delta I=I_{\eta \eta }+I_{\xi \xi }$. Furthermore, if we rewrite this equality with the directions $\eta \left(x\right)=N\left(x\right)=\frac{\nabla I\left(x\right)}{\left|\nabla I\left(x\right)\right|}$, where |∇I(x)|≠0, and ξ(x)=T(x) with T(x)·N(x)=0 and |T(x)|=1, then $\Delta I=I_{NN}+I_{TT}$, where $I_{NN}$ and $I_{TT}$ are the second derivatives of image I in the N-direction and T-direction respectively. The isotropy means that the diffusion is equivalent in the two directions. Thus, most of the diffusion operators can be decomposed as a weighted sum of $I_{NN}$ and $I_{TT}$ [55]. With the usual notation $I_{x}$, $I_{y}$, $I_{xx}$, $I_{xy}$, $I_{yy}$ for the first and second partial derivatives of *I*, $I_{NN}$ and $I_{TT}$ can be expressed as
(9)$$I_{NN}=\frac{1}{{\left|\nabla I\right|}^{2}}\left(I_{x}^{2}I_{xx}+I_{y}^{2}I_{yy}+2I_{x}I_{y}I_{xy}\right)$$
and
(10)$$I_{TT}=\frac{1}{{\left|\nabla I\right|}^{2}}\left(I_{x}^{2}I_{yy}+I_{y}^{2}I_{xx}-2I_{x}I_{y}I_{xy}\right).$$

Then Equation (3) can be rewritten as
(11)$$\{\begin{array}{l} \frac{\partial I}{\partial t}=c_{1}\left({\left|\nabla I\right|}^{2}\right)I_{TT}+c_{2}\left({\left|\nabla I\right|}^{2}\right)I_{NN}\;\;on\;\;\Omega \times \left(0,T\right)\\ \frac{\partial I}{\partial N}=0\;\;on\;\partial \Omega \times \left(0,T\right)\\ I\left(x,0\right)=I_{0}\left(x\right)\;on\;\Omega \end{array}$$
where $c_{1}\left({\left|\nabla I\right|}^{2}\right)=\frac{1}{{\left(1+{\left|\nabla I\right|}^{2}\right)}^{\left(1-\beta \right)/2}}$, $c_{2}\left({\left|\nabla I\right|}^{2}\right)=2{\left|\nabla I\right|}^{2}\cdot \frac{1-\beta }{2}\cdot \frac{1}{{\left(1+{\left|\nabla I\right|}^{2}\right)}^{\frac{\left(1-\beta \right)}{2}+1}}$. Here the diffusion operators can be regarded as a weighted sum of $I_{NN}$ and $I_{TT}$. Combining the gray level indicator function constructed previously, we propose the following new anisotropic diffusion equation:(12)$$\{\begin{array}{l} \frac{\partial I}{\partial t}=b\left(I\right)\left[c_{1}\left({\left|\nabla I\right|}^{2}\right)I_{TT}+c_{2}\left({\left|\nabla I\right|}^{2}\right)I_{NN}\right]\;on\;\;\Omega \times \left(0,T\right)\\ \frac{\partial I}{\partial N}=0\;\;on\;\partial \Omega \times \left(0,T\right)\\ I\left(x,0\right)=I_{0}\left(x\right)\;on\;\Omega \end{array}$$

For different regions of the image, we hope to implement isotropic diffusion in the homogeneous region, and anisotropic diffusion in the edge and local detail region. Meanwhile, diffusion spreads along the tangent direction at the edge, rather than diffuse along the gradient direction. Hence, in further designing a diffusion coefficient that meets the requirements, Equation (9) can be rewritten as follows:(13)$$\{\begin{array}{l} \frac{\partial I}{\partial t}=b_{1}\left(I\right)c_{1}\left({\left|\nabla I\right|}^{2}\right)I_{TT}+b_{2}\left(I\right)c_{2}\left({\left|\nabla I\right|}^{2}\right)I_{NN}\;on\;\;\Omega \times \left(0,T\right)\\ \frac{\partial I}{\partial N}=0\;\;on\;\partial \Omega \times \left(0,T\right)\\ I\left(x,0\right)=I_{0}\left(x\right)\;on\;\Omega \end{array}$$
where $b_{1}\left(I\right)=b\left(I\right)$, $b_{2}\left(I\right)=\theta b\left(I\right)$. Parameter θ=0 at the edge and θ=1 at the non-edge. In this way, the diffusion at the edge can be guaranteed to occur, and the diffusion occurs only along the tangent direction and not along the gradient direction.

## 4. Experimental Scheme

By replacing the derivatives with finite differences, we have following scheme:(14)$$U_{x}\left(I_{i,j}\right)=I_{i+1,j}-I_{i,j},\;U_{y}\left(I_{i,j}\right)=I_{i,j+1}-I_{i,j},$$
(15)$$U_{xx}\left(I_{i,j}\right)=I_{i+1,j}+I_{i-1,j}-2I_{i,j},\;U_{xy}\left(I_{i,j}\right)=I_{i+1,j+1}+I_{i,j}-I_{i+1,j}-I_{i,j+1},$$
(16)$$U_{yy}\left(I_{i,j}\right)=I_{i,j+1}+I_{i,j-1}-2I_{i,j}.$$
where the space step h=1.

Then, we adopt the iterative method as follows:(17)$$b_{i,j}=\frac{2{\left|\sqrt{I_{i,j}}\sigma \right|}^{\alpha }}{M^{\alpha }+{\left|\sqrt{I_{i,j}}\sigma \right|}^{\alpha }},$$
(18)$$D_{i,j}^{n}={\left(U_{x}\left(I_{i,j}^{n}\right)\right)}^{2}+{\left(U_{y}\left(I_{i,j}^{n}\right)\right)}^{2},$$
(19)$$R_{i,j}^{n}=1+{\left(D_{i,j}^{n}\right)}^{\frac{1-\beta }{2}},$$
(20)$$U_{NN}\left(I_{i,j}^{n}\right)=\frac{1}{D_{i,j}^{n}}\left({\left(U_{x}\left(I_{i,j}^{n}\right)\right)}^{2}U_{xx}\left(I_{i,j}^{n}\right)+{\left(U_{y}\left(I_{i,j}^{n}\right)\right)}^{2}U_{yy}\left(I_{i,j}^{n}\right)+2U_{x}\left(I_{i,j}^{n}\right)U_{y}\left(I_{i,j}^{n}\right)U_{xy}\left(I_{i,j}^{n}\right)\right),$$
(21)$$U_{TT}\left(I_{i,j}^{n}\right)=\frac{1}{D_{i,j}^{n}}\left({\left(U_{x}\left(I_{i,j}^{n}\right)\right)}^{2}U_{yy}\left(I_{i,j}^{n}\right)+{\left(U_{y}\left(I_{i,j}^{n}\right)\right)}^{2}U_{xx}\left(I_{i,j}^{n}\right)-2U_{x}\left(I_{i,j}^{n}\right)U_{y}\left(I_{i,j}^{n}\right)U_{xy}\left(I_{i,j}^{n}\right)\right)$$
(22)$$I_{i,j}^{n+1}=I_{i,j}^{n}+\gamma \left(\frac{b_{i,j}}{R_{i,j}^{n}}U_{TT}\left(I_{i,j}^{n}\right)+\frac{\theta b_{i,j}}{R_{i,j}^{n}}U_{TT}\left(I_{i,j}^{n}\right)\right)$$
(23)$$I_{i,j}^{0}=I_{0}=I\left(ih,jh\right)$$
(24)0≤i≤P, 0≤j≤Q,
(25)$$I_{i,0}^{n}=I_{i,1}^{n},\;I_{0,j}^{n}=I_{1,j}^{n},$$
(26)$$I_{P,j}^{n}=I_{P-1,j}^{n},\;I_{i,Q}^{n}=I_{i,Q-1}^{n}.$$
where γ is the time step for n=1,2⋯.

## 5. Numerical Experiments and Discussion

This section is devoted to demonstrating the ability of our method to deal with multiplicative speckle noise. We carried out experiments on synthetic images, real ultrasound images and RGB color images by comparing different methods. The qualitative and quantitative performance analysis is given in the synthetic images tests, the feasibility and effectiveness of the proposed model in practical application is demonstrated in the real ultrasound images tests, and the practicability of the proposed method is shown in the RGB color images tests. All the experiments are performed under Windows 10 and MATLAB R2015b, running on a laptop with an Intel(R) Pentium(R) CPU (2.16 GHz) and 4 GB memory.

### 5.1. Synthetic Image Experiments

In this sub-section, the capability of the proposed models has been evaluated with comparisons to the representative methods: SRAD method code can be found at (http://viva.ee.virginia.edu/downloads.html). Optimized Bayesian non local means (OBNLM) method [56] code can be obtained from Coupe’s personal website (https://sites.google.com/site/pierrickcoupe/softwares/denoising-for-medical-imaging/speckle-reduction/obnlm-package). The MATLAB code for ADMSS method can be obtained at (https://www.mathworks.com/matlabcentral/fileexchange/52988-anisotropic-diffusion-with-memory-based-on-speckle-statistics-for-ultrosound-images). Since our model is inspired by the DDND model, it is necessary for us to make a comparison. The parameter settings used for each method are parameters in the original code. We carried out two experiments with synthetic images. The first one was intended to show the performance of removing speckle noise and preserving details by using Equation (8), in which the method of Equations (14)–(19), (23)–(26) is adopted. Whereas the second experiment shows the capability of preserving structures by using Equation (13), in which the method of Equations (14)–(26) is adopted. In addition to the real ultrasound images, for each noise image in the experiment, a noisy observation is produced according to the speckle noise Model (2) with the levels of standard deviations *σ*.

To evaluate the denoising effect, we use the peak signal-to-noise ratio (PSNR) [19] and the mean absolute deviation error (MAE) [19] as the measurement indexes, which are defined as follows: (27)$$\mathrm{PSNR}:=10\log_{10}\frac{MN{\left|\max I_{O}-\min I_{O}\right|}^{2}}{{\left\|I-I_{O}\right\|}_{L^{2}}^{2}}\left(\mathrm{dB}\right),$$
(28)$$\mathrm{MAE}:=\frac{{\left\|I-I_{O}\right\|}_{L^{1}}}{MN}.$$
where I and $I_{O}$ represent the denoising image and original image, respectively, M and N are the size of the image. A high PSNR and a low MSE of the image show good performance of the model. In addition, the structural similarity (SSIM) [57] is introduced to evaluate the similarity between I- and -$I_{O}$. The stop condition guidelines serve to achieve optimal PSNR, MAE, and SSIM.

During the experimental testing of our model, there are two important parameters (α, β) to be tuned. The parameters and the stopping criterion of all methods were tuned mutually to achieve the maximal PSNR or the best MAE and SSIM for a fair comparison. As we can see in Table 1, we found that 1≤α≤2 and 0≤β≤0.3 can provide optimal results of our method. At the same time, the optimal result for the noisy image changes slightly when the parameters α and β change in this range.

We first used Model (8) to test two synthetic images in Figure 3: Shepp-Logan head phantom (256 × 256 pixels) (the Shepp-Logan head phantom image is available at http://bigwww.epfl.ch/demo/ctreconsturction/example.html) and Cameraman (256 × 256 pixels) (Cameraman image is available at (http://sipi.usc.edu/services/data-base/index.html)). The synthetic images were corrupted with different levels of noise, three levels of noise were tested by setting σ={1, 2, 3}. Figure 4 shows the denoising results for the Shepp-Logan head phantom image corrupted by speckle noise with σ=3 and Figure 5 shows the denoising results for the Cameraman image corrupted by speckle noise with σ=2. Because the noise is generated randomly according to the standard deviation, as illustrated in Figure 4 and Figure 5, the effects of SRAD and ADMSS are not particularly ideal. SRAD’s result suffers from the blurring structure, and the restored images from SRAD and ADMSS tend to produce some isolated white spots. OBNLM and DDND are slightly better. OBNLM looks for similar patches in the image, takes of large homogeneous and reduces the speckle. However, the oversmoothing seriously harms the structural detail. DDND is good at denoising, but the structure is slightly fuzzy. We can also see that SRAD, OBNLM and ADMSS fails to preserve the details in the low gray level texture structures (the three gray little ovals in Figure 4 and the building in the lower right corner of Figure 5), while DDND and our method are better because of the guidance of the region indicator. Compared with them, our method is able to obtain the best results in the visual aspect. Consequently, two examples verify that the proposed Model (8) produces the best results. To quantify the Model (8) performance, Table 2 shows comparisons of our method with other denoising methods on the basis of the PSNR, MAE and SSIM values. It can be revealed that our method provides the optimal PSNR, MAE and SSIM values in all examples, which are shown in bold face in Table 2. These results further confirm that the proposed method outperforms its competitors.

In addition, we used Model (13) to test the performance of preserving structures on the Lena image (256 × 256 pixels) (the Lena image is available at (http://sipi.usc.edu/services/data-base/index.html)), the results are shown in Figure 6. To more clearly represent the ability of preserving structures, we zoom in the local details of the Lena image for comparison in Figure 7. By visual inspection, SRAD retains isolated noise points, making it difficult to keep sharp edges. For OBNLM, it can be seen that there exist gray value ‘drifts’. ADMSS generates few white pixels. At the same time, a common feature of SRAD and ADMSS methods is the failure to preserve the detailed structure (Lena’s hair in Figure 7). DDND and our method have no such problem. As can be seen in Figure 7, the edge preservation effect of Model (13) is significantly better than that of other methods. Compared with them, our result is better at protecting finer edge structures and restoring faint geometrical structures of the images except for it removes the speckle noise effectively.

### 5.2. Real Ultrasound Image Denoising Experiment

Our main purpose is to apply denoising model (8) to remove multiplicative speckle noise, which usually occurs in real-world images, such as the speckle noise in ultrasound images. Therefore, we present the experiment results to illustrate the denoising capability of the proposed model for real ultrasound images in this paragraph. We took the PSNR as the stopping criterion in the previous synthetic images experiment, but there was no ground truth in the real image experiment, so we conducted the experiment after doing some new work. For the sake of brevity, we will only show the results obtained through visual optimization.

#### 5.2.1. The Innovation of Our Work

To reflect the denoising ability of our model to the real ultrasound images, we have done the following work: The iteration stopping criteria are given: We used PSNR as the stop criterion in the experiment with the synthetic image, but there was no ground truth in the real image experiment, so we gave two new stop criteria: One was that we observed whether the noise part $n=\frac{I_{0}-I}{\sqrt{I}}$ was close to Gaussian, which can be measured by the kurtosis k(x), which is a fourth-order statistic [58],
(29)$$k\left(x\right)=\frac{\sum {\left(x_{i}-\mu \right)}^{4}}{\frac{n}{\sigma^{4}}}-3.$$
where μ is the mean and σ is the standard deviation. The kurtosis should be zero when the noise part is Gaussian. Thus, we stop iteration when the kurtosis falls below 0.001. The other was to judge the independence of image *I* and noise *n*, where we use correlation as a standard. When the correlation between two variables is small, we stop the iteration.Parameter training: Our parameters (α, β) are robust to noise standard deviation, so the exact optimal value can be obtained by learning. First, we divide the same class of images in the USC-SIPI image database into two parts: training image and test image. By using the original image and the noise images with different variances in the training image, the optimal parameter values are obtained by using regression after experiments. We found the optimal parameter values of α=2 and β=0.04. With the above model, algorithm and optimal parameters, the test image can be checked, and the PSNR indexes can be calculated through the denoising images and the original images.Deviation correction: To improve the denoising effect, we can carry out post-processing. Using the output results, we can calculate: $n=\frac{I_{0}-I}{\sqrt{I}}$. If we use the first stop criterion, the kurtosis of n is 0. However, its histogram is not necessarily Gaussian, it should be the distribution of the noise we know. So we do the histogram specification of *n* to bring the adjusted $\hat{n}$ closer to the required Gaussian distribution.Post-processing: We use the $\hat{n}$ obtained above to solve the quadratic equation $I_{0}=I+\sqrt{I}\hat{n}$ to get $\hat{I}$.

#### 5.2.2. Experimental Results

In this subsection, we present the visual comparison of denoising results on three real ultrasound images: Liver image (Figure 8) (Original image is available at http://www.ultrasoundcases.info/Cases-Home.aspx), Ovary Cancer image (Figure 9) (original image is available in the work of Zhou et al. (2017)) and Ultrasound image (Figure 10) (Original image is available at http://telin.ugent.be/~sanja/Sanja_files/UltrasoundDemo.htm). The results of different denoising methods are shown in Figure 8, Figure 9 and Figure 10. Visually, we can easily observe that SRAD shows oversmoothing, resulting in the desired information being unable to be preserved, and OBNLM has the vague effect, one can see there exist gray value ‘drifts’ in the denoising results, which is undesirable. Although ADMSS, which uses a memory element, preserves the speckle, it generates a few white pixels in large homogeneous background regions. Three methods will seriously affect the diagnosis process. DDND and our method do a better job preserving small details than SRAD, OBNLM and ADMSS, especially the area marked by a red rectangle in Figure 8, Figure 9 and Figure 10, which may be an object of interest to potential clinical diagnosis. In light of these visual results, we can conclude that the proposed method not only effectively removes the speckle noise but also better preserves relevant structures that might be useful for diagnostic purpose.

### 5.3. RGB Color Image Denoising Experiment

In this subsection, we present the experiment results to illustrate the denoising capability of the proposed models for RGB color image. We used Model (7) to test on lena_color (512 × 512 pixels), mandril_color (512 × 512 pixels), airplane_color (512 × 512 pixels), peppers_color (512 × 512 pixels), house_color (256 × 256 pixels) and butterfly_color (256 × 256 pixels) (all images are available at (http://sipi.usc.edu/services/data-base/index.html)). The results are shown in Figure 11. For the sake of compactness, we will only show the results obtained through visual optimization. The visual results show that our method is effectively able to remove speckle noise and retain small details, and further confirm that the proposed model is also suitable for RGB color images denoising.

## 6. Conclusions

This paper presents a novel method to remove the noise in the multiplication involved image processing system. Experiments were carried out on synthetic, real ultrasound and RGB color images. During the experiments, quantitative measures were used to compare several denoising methods in the synthetic experiments, quantitative measures were used to compare several denoising methods in synthetic images, kurtosis and correlation as iteration stopping criteria for the lack of ground truth in real images. In addition, the proposed method is also suitable for RGB color image denoising. Experiments show that the performance of our method outperforms the other methods.

Although the proposed approach achieves better performance than several existing approaches, we will continue to investigate the proposed method with better strong edge preserving and denoising capabilities.

## Figures and Tables

**Figure 1 sensors-19-03164-f001:**
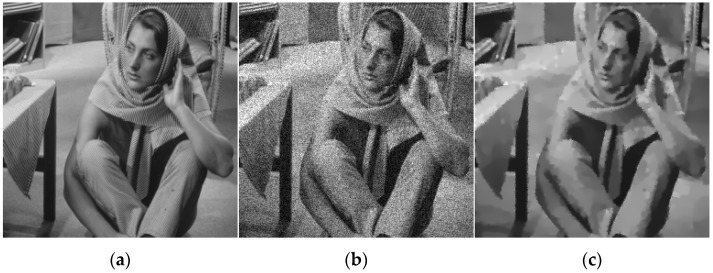
Denoising result of the PM model. (**a**) Original image; (**b**) noisy image; (**c**) denoising result of the PM model.

**Figure 2 sensors-19-03164-f002:**
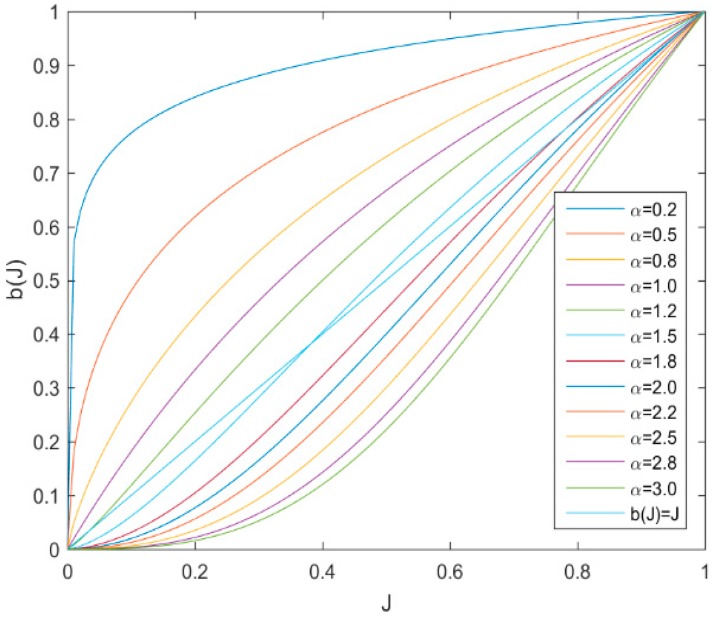
Influence of the parameter α.

**Figure 3 sensors-19-03164-f003:**
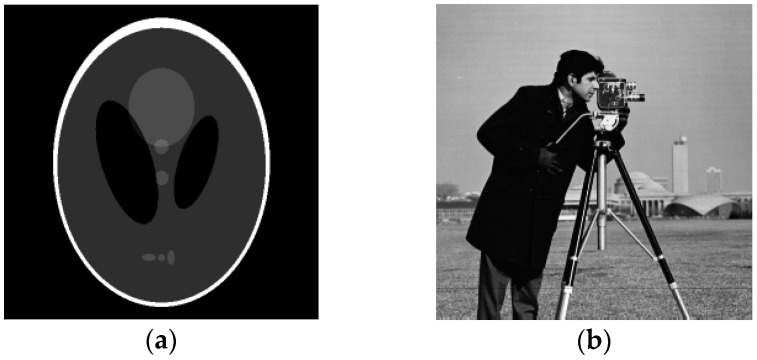
The synthetic test image: (**a**) Shepp-Logan head phantom (256 × 256 pixels); (**b**) Cameraman (256 × 256 pixels).

**Figure 4 sensors-19-03164-f004:**
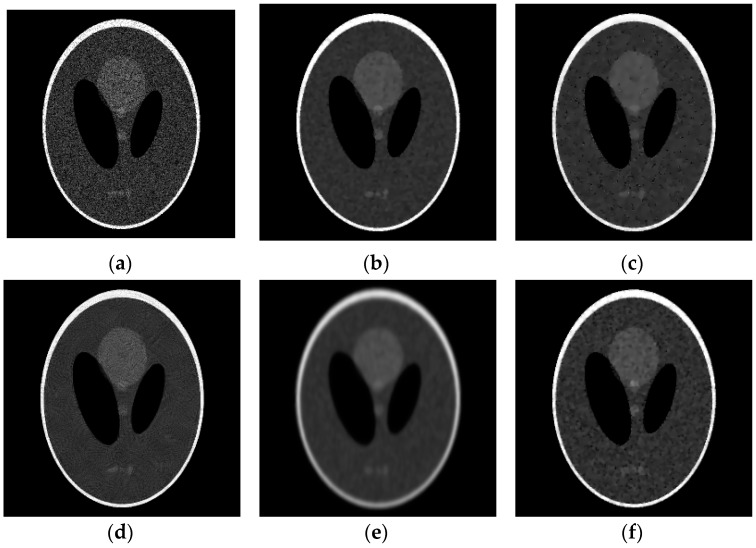
Results obtained with different denoising methods applied to the Shepp-Logan head phantom image corrupted with noise (σ = 3): (**a**) noisy; (**b**) ours; (**c**) SRAD; (**d**) OBNLM; (**e**) ADMSS; (**f**) DDND.

**Figure 5 sensors-19-03164-f005:**
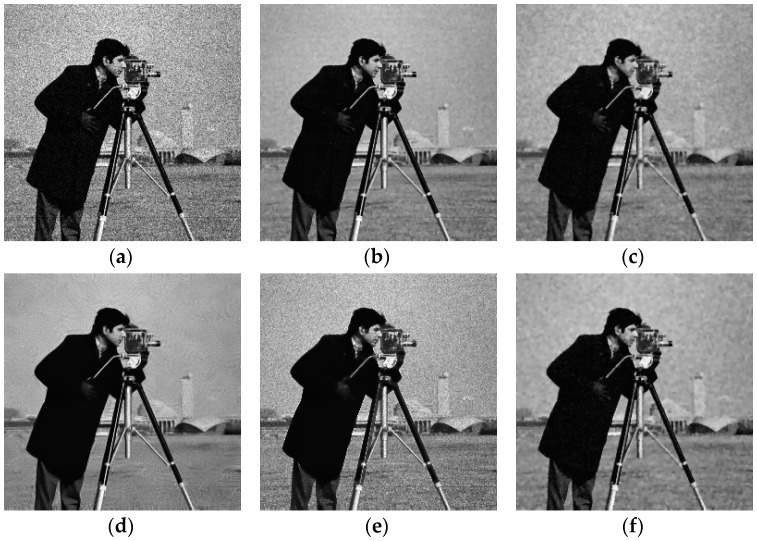
Results obtained with different denoising methods applied to the Cameraman image corrupted with noise (σ = 2): (**a**) noisy; (**b**) ours; (**c**) SRAD; (**d**) OBNLM; (**e**) ADMSS; (**f**) DDND.

**Figure 6 sensors-19-03164-f006:**
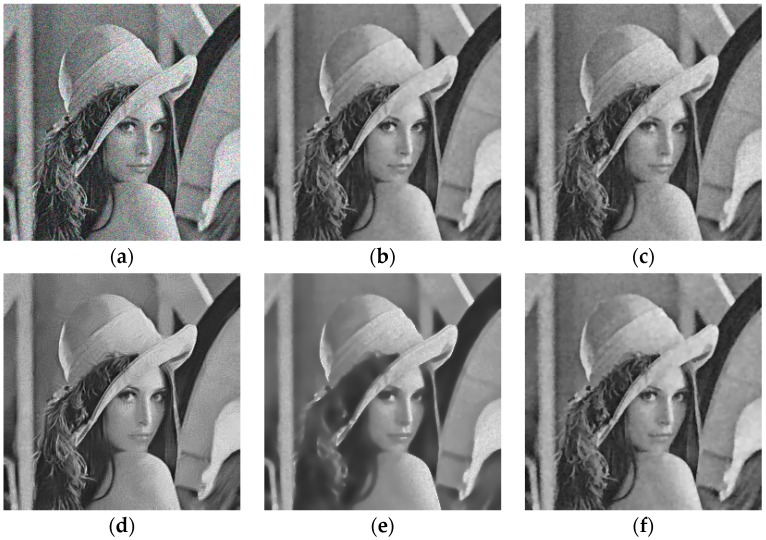
Results obtained with different denoising methods applied to Lena image corrupted with noise (σ = 2): (**a**) noisy; (**b**) ours; (**c**) SRAD; (**d**) OBNLM; (**e**) ADMSS; (**f**) DDND.

**Figure 7 sensors-19-03164-f007:**
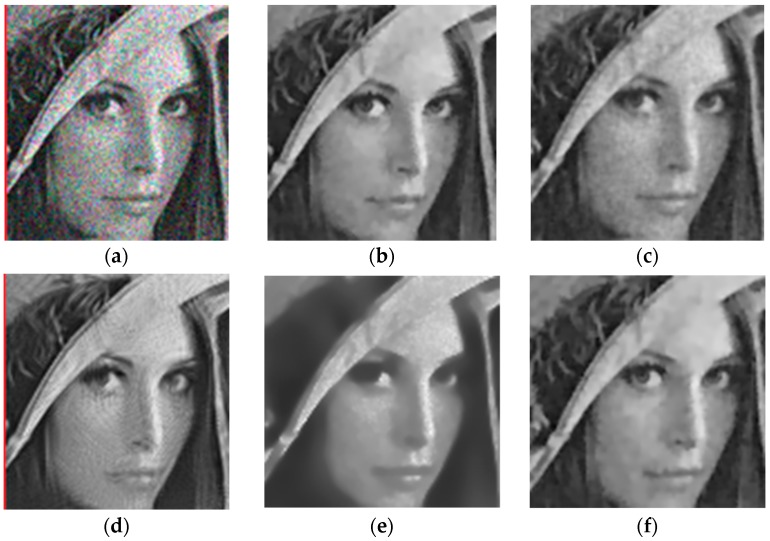
Close-up of results obtained with different denoising methods applied to the Lena image corrupted with noise (σ = 2): (**a**) noisy; (**b**) ours; (**c**) SRAD; (**d**) OBNLM; (**e**) ADMSS; (**f**) DDND.

**Figure 8 sensors-19-03164-f008:**
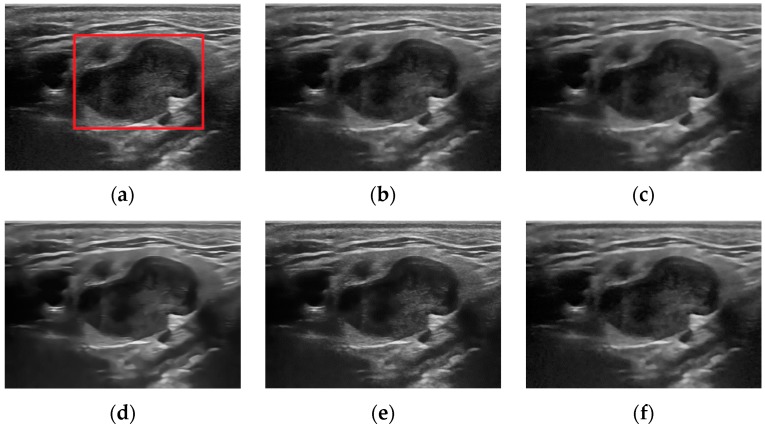
Results obtained with different denoising methods applied to the Liver ultrasound image: (**a**) original; (**b**) ours; (**c**) SRAD; (**d**) OBNLM; (**e**) ADMSS; (**f**) DDND.

**Figure 9 sensors-19-03164-f009:**
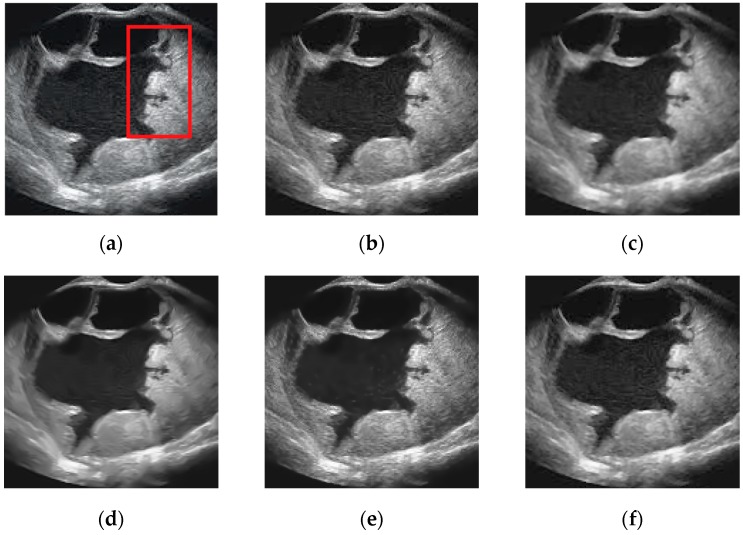
Results obtained with different denoising methods applied to the Ovary Cancer ultrasound image: (**a**) original; (**b**) ours; (**c**) SRAD; (**d**) OBNLM; (**e**) ADMSS; (**f**) DDND.

**Figure 10 sensors-19-03164-f010:**
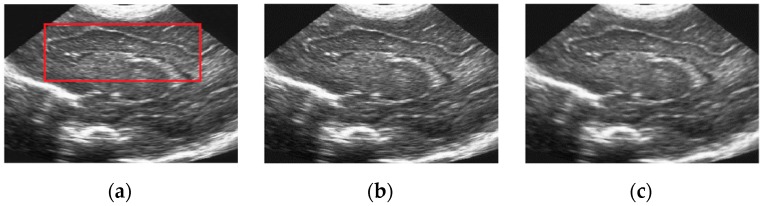
Results obtained with different denoising methods applied to the ultrasound image: (**a**) original; (**b**) ours; (**c**) SRAD; (**d**) OBNLM; (**e**) ADMSS; (**f**) DDND.

**Figure 11 sensors-19-03164-f011:**
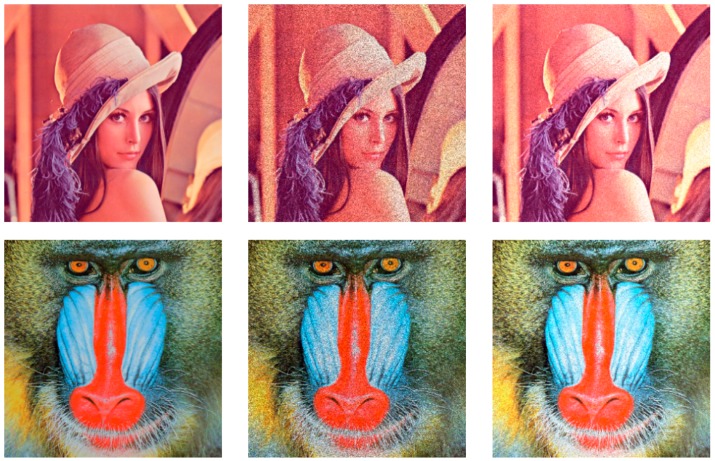
Denoising results of the RGB color image: (**a**) original; (**b**) noisy; (**c**) results.

**Table 1 sensors-19-03164-t001:** Performance of parameters in our model.

	The Shepp-Logan Head Phantom Image			The Cameraman Image		
	(σ=1)			(σ=1)		
Parameters	PSNR	MAE	SSIM	PSNR	MAE	SSIM
α=0.5,β=0.2	38.10	1.21	0.98	31.26	4.68	0.85
α=0.5,β=0.4	37.18	1.43	0.97	30.99	4.85	0.86
α=0.5,β=0.6	36.18	1.73	0.96	30.69	5.08	0.85
α=0.5,β=0.8	34.82	2.10	0.93	30.20	5.43	0.84
α=1.0,β=0.1	38.52	1.14	0.99	31.44	4.09	0.86
α=1.0,β=0.2	37.96	1.23	0.98	31.31	4.59	0.83
α=1.0,β=0.3	37.49	1.34	0.97	31.15	4.76	0.87
α=1.0,β=0.4	37.08	1.49	0.97	31.10	4.79	0.85
α=1.5,β=0.1	38.24	1.17	0.99	31.35	4.57	0.87
α=1.5,β=0.2	37.88	1.28	0.99	31.30	4.63	0.87
α=1.5,β=0.3	37.30	1.41	0.99	31.27	4.65	0.87
α=2.0,β=0.1	38.03	1.24	0.99	31.35	4.59	0.87
α=2.0,β=0.2	37.64	1.35	0.99	31.31	4.65	0.87
α=2.5,β=0.2	37.01	1.48	0.99	31.28	4.66	0.87

**Table 2 sensors-19-03164-t002:** PSNR, MAE and SSIM.

	PSNR			MAE			SSIM		
σ	1	2	3	1	2	3	1	2	3
**The Shepp-Logan Head Phantom Image**									
SRAD	35.32	32.00	29.69	1.80	2.54	3.18	0.93	0.92	0.91
OBNLM	37.80	31.91	30.04	1.46	2.39	3.56	0.97	0.93	0.85
ADMSS	37.58	32.44	30.31	1.19	2.43	3.50	0.98	0.93	0.92
DDND	37.61	32.87	30.03	1.42	2.37	3.39	0.97	0.94	0.93
Ours	**38.52**	**33.55**	**30.92**	**1.14**	**2.06**	**2.93**	**0.99**	**0.96**	**0.94**
**The Cameraman Image**									
SRAD	30.62	27.45	25.38	5.12	6.91	9.81	0.82	0.73	0.67
OBNLM	30.96	27.48	23.25	4.15	6.79	12.31	0.88	0.74	0.52
ADMSS	31.32	27.49	25.41	4.39	6.80	8.84	0.88	0.75	0.67
DDND	31.39	27.55	25.58	4.64	6.75	8.35	0.86	0.75	0.69
Ours	**31.44**	**27.60**	**25.69**	**4.09**	**6.63**	**8.13**	**0.86**	**0.77**	**0.72**

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
