# Peer review of "Anisotropic Diffusion Based Multiplicative Speckle Noise Removal"

_sensors, 2019, doi:10.3390/s19143164_

Round 1

Reviewer 1 Report

1.  An anisotropic diffusion model based on image statistics that, includes information on the gradient of the image, gray levels, and noise standard deviation of the image, is proposed.

2. The iteration stopping criteria based on kurtosis and correlation in view of the lack of ground truth in real image experiments, is proposed.

3.  The optimal values of the parameters in the model are obtained by learning.

4. The simulation results show that the proposed model can effectively remove the speckle noise and retain minute details of the images for the real ultrasound and RGB color images.

5. In line 39, ”Zhou et al. (2015, 2017)” should be revised to “Zhou et al. [41-42]”.

6. In line 82, ”Zhou et al. (2015)” should be revised to “Zhou et al. [41]”.

7. In line 35, ”Zhou et al. (2017)” should be revised to “Zhou et al. [42]”.

Author Response

Dear Reviewer:

Thank you for your comments concerning our manuscript entitled “Anisotropic Diffusion Based Multiplicative Speckle Noise Removal” (ID: sensors-539493). Those comments are all valuable and helpful for revising and improving our paper, as well as the important guiding significance to our researches. We have studied comments carefully and have made correction which we hope meet with approval. Revised portion are marked in red in the paper. The main corrections in the paper and the responds to your comments are as following:

1. Considering the your suggestion, we have revised line 14-16: An anisotropic diffusion model based on image statistics that, includes information on the gradient of the image, gray levels, and noise standard deviation of the image, is proposed.

2. Considering the your suggestion, we have revised line 19-21: The iteration stopping criteria based on kurtosis and correlation in view of the lack of ground truth in real image experiments, is proposed.

3. Considering the your suggestion, we have revised line 21: The optimal values of the parameters in the model are obtained by learning.

4. Considering the your suggestion, we have revised line 22-24: The simulation results show that the proposed model can effectively remove the speckle noise and retain minute details of the images for the real ultrasound and RGB color images.

5. In line 38: ”Zhou et al. (2015, 2017)” were revised to “Zhou et al. [41-42]”.

6. In line 81, ”Zhou et al. (2015)” were revised to “Zhou et al. [41]”.

7. In line 84, ”Zhou et al. (2017)” were revised to “Zhou et al. [42]”.

We appreciate for your warm work earnestly, and hope that the correction will meet withapproval.

Once again, thank you very much for your comments and suggestions.

Yours sincerely,

Mei Gao

Name: Baosheng Kang

Reviewer 2 Report

This paper presents a novel method to remove the noise in the multiplication involved image processing system. 

Key strength 

The problem is well presented and modeled 

Detailed mathematical analysis is provided 

Rigorous research is thoroughly provided 

Simulation results are superior 

Author Response

Dear Reviewer:

Thank you for your comments concerning our manuscript entitled “Anisotropic Diffusion Based Multiplicative Speckle Noise Removal” (ID: sensors-539493). Those comments are all valuable and helpful for revising and improving our paper, as well as the important guiding significance to our researches. We have studied comments carefully and have made correction which we hope meet with approval. Revised portion are marked in red in the paper.

Considering your comments, We have re-written the beginning of the conclusion.

Line 376: This paper presents a novel method to remove the noise in the multiplication involved image processing system. 

We appreciate for your warm work earnestly, and hope that the correction will meet with approval.

Once again, thank you very much for your comments and suggestions.

Yours sincerely,

Mei Gao

Name: Baosheng Kang

Reviewer 3 Report

Referee report on ” Anisotropic Diffusion Based Multiplicative Speckle Noise Removal” by Gao et al.

This is a well-written paper putting forward a new and superior method to deal with speckle noise. The paper is a significant advance in the field. The title is appropriate, the abstract presents a good summary, the use of figures is effective, the mathematics is clearly presented, the references are appropriate. In my opinion, the paper is already ready for publication after the authors have fixed some minor bugs. The only thing I missed is that the authors provide a very brief discussion on physical origins of speckle noise. It would add to the coherence and completeness of this very good paper.

The list of minor bugs I have found is (even though this list is long, the paper is else clearly and well written):

Line 13: “of a compressed”

Line 19: “along the boundaries’ direction rather than along the vertical direction of the boundaries,” should be “along the boundaries rather than perpendicular to the boundaries,”

Line 23: “…effect, post-processing…”

Line 63: This sentence is unclear to me. (“so we need to …”)

Line 64: find and replace “denosing” -> “denoising” in the entire manuscript. At least 15 times the erroneous “denosing” appear, e.g. in many figure captions

Line 72: “…while it preserves…”

Line 98: “…given: kurtosis”

Line 99: “…and correlation…”¨

Line 104: “…Gaussian. Therefore,…”

Line 107: “…introduce the PM model…”

Line 108: “…we propose an anisotropic…”

Line 109: “Our experimental…”

Line 110: “ultrasound” with small u

Line 111: “…conclusion is presented…”

Line 124: “…edge. Thus,…”

Line 146: “…the grayscale of the original image…”

Line 167: “b(I) approaches 1”

Line 168: “near the boundaries of the region…”

Line 176: “we got a constant…”

Line 183: “…boundaries of the region…”

Line 185: “…boundaries of the region…”

Line 186: : “…boundaries of the region…”

Line 186: “along the boundaries rather than…”

Line 237: “researches” -> “tests”

Line 238: “researches” -> “tests”

Line 239: “researches” -> “tests”

Line 249: “…performance of removing speckle noise and preserving details by using equation (7)” Observe that there are 3 corrections here.

Line 250: “by using equation (10)”

Line 266: “MAE and SSIM for  a fair comparison…”

Line 267: “…can provide optimal…”

Line 268: “…for the noisy image changes slightly…”??

Line 272: “…shows the denoising results…”

Line 279: “denoising”

Line 282: “denoising”

Table 2: The second major column of table 2 says MSE, but elsewhere in the text MAE appears. I believe that this is the same, and the authors should make the acronyms consistent.

Line 293: “denoising”

Line 295: “denoising”

Line 298: MSE vs MAE?

Line 298: “…the our method…”

Line 299: MSE vs MAE?

Line 301: the ‘1’ should be upper case

Line 303: “…we zoom in the….”

Line 304: “…points…” ??

Line 307: “…preserve the detailed structure…”

Line 310: “…except for it removes…”

 Line 313: “denoising”

Line 320: Could the authors find a more descriptive subtitle heading which indicates the content of the subsection.

Line 326: “…by the kurtosis…”

Line 326: “…is a fourth –order…”

Line 329: “…kurtosis is below….” . Could you here provide your exact stopping criterion. I would guess the the kurtosis is below some very small value.

Line 338: “indexes”

Line 342: “it should be…”

Line 348: “denoising”

Line 350: “denoising”

Line 352: “denoising”

Line 348: “denosing”

Line 359: It is not clear what “the vague effect is”. Could you elaborate on this?

Line 363: “…object of interest to potential…”

Line 373: “…that our method…”

Line 405 “denoising”

Line 405: This should be figure 11, rather then figure 1.

Author Response

Dear Reviewer:

Thank you for your comments concerning our manuscript entitled “Anisotropic Diffusion Based Multiplicative Speckle Noise Removal” (ID: sensors-539493). Those comments are all valuable and helpful for revising and improving our paper, as well as the important guiding significance to our researches. We have studied comments carefully and have made correction which we hope meet with approval. Revised portion are marked in red in the paper. The main corrections in the paper and the responds to your comments are as following:

1. Line 13: the statements of “of compressed” were corrected as “of a compressed”

2. Line 19: the statements of “along the boundaries’ direction rather than along the vertical direction of the boundaries,” were corrected as “along the boundaries rather than perpendicular to the boundaries,”

3. Line 23: the statements of “…effect, the post-processing…” were corrected as “…effect, post-processing…”

4. Response to comment: Line 63: This sentence is unclear to me. (“so we need to …”)

Response: Line 63 the statements of “so we need to denoising by the way of adding noise.” were revised as “we can add noise to the original image in the synthetic image experiment, and then denoising.”

5. Line 64: replace “denosing” -> “denoising”

6. Line 72: the statements of “…while preserves…” were corrected as “…while it preserves…”

7. Line 98: the statements of “…given: the kurtosis” were corrected as “…given: kurtosis”

8. Line 99: the statements of “…and the correlation…” were corrected as “…and correlation…”

9. Line 104: the statements of “…Gaussian, therefore…” were corrected as “…Gaussian. Therefore,…”

10. Line 107: the statements of “…introduce PM model…” were corrected as “…introduce the PM model…”

11. Line 108: the statements of “…we propose anisotropic…” were corrected as “…we propose an anisotropic…”

12. Line 109: the statements of “Experimental…” were corrected as “Our experimental…”

13. Line 110: replace “Ultrasound” -> “ultrasound”

14. Line 111: the statements of “…conclusion is give presented…” were corrected as “…conclusion is presented…”

15. Line 124: the statements of “…edge, thus…” were corrected as “…edge. Thus,…”

16. Line 146: the statements of “…the original image’s grayscale…” were corrected as “…the grayscale of the original image…”

17. Line 167: the statements of “... closes to 1” were corrected as “

 ...approaches 1”

18. Line 168: the statements of “near the region’s boundaries…” were corrected as “near the boundaries of the region…”

19. Line 176: the statements of “we got an constant…” were corrected as “we got a constant…”

20. Line 183: the statements of “…the region’s boundaries…” were corrected as “…boundaries of the region…”

21. Line 185: the statements of “…the region’s boundaries…” were corrected as “…boundaries of the region…”

22. Line 186: the statements of “…the region’s boundaries…” were corrected as “…boundaries of the region…”

23. Line 186: the statements of “…along the boundaries’ direction rather than…” were corrected as “along the boundaries rather than…”

24. Line 237: replace “researches” -> “tests”

25. Line 238: replace “researches” -> “tests”

26. Line 239: replace “researches” -> “tests”

27. Line 249: the statements of “…performance of remove speckle noise and preserve details by used equation (7)” were corrected as “…performance of removing speckle noise and preserving details by using equation (7)”

28. Line 250: the statements of “by use equation (10)” were corrected as “by using equation (10)”

29. Line 266: the statements of “MAE and SSIM in order to fair comparison” were corrected as “MAE and SSIM for a fair comparison”

30. Line 267: the statements of “…can provided optimal…” were corrected as “…can provide optimal…”

31. Line 268: the statements of “…for noisy image slightly…” were corrected as “…for the noisy image changes slightly…”

32. Line 272: the statements of “…shows the for the denoising results…” were corrected as “…shows the denoising results…”

33. Line 279: replace “denosing” -> “denoising”

34. Line 282: replace “denosing” -> “denoising”

35. Response to comment: (Line Table 2: The second major column of table 2 says MSE, but elsewhere in the text MAE appears. I believe that this is the same, and the authors should make the acronyms consistent.)

Response: We are very sorry for our negligence of this work, the standard used in this article is MAE.

36. Line 293: replace “denosing” -> “denoising”

37. Line 295: replace “denosing” -> “denoising”

38. Response to comment: (Line 298: MSE vs MAE?)

Response: MAE

39. Line 301: modigy the ‘1’ to be upper case

40. Line 303: the statements of “…we zooming the…” were corrected as “…we zoom in the….”

41. Line 304: the statements of “…point…” were corrected as “… points…”

42. Line 307: the statements of “…preserve the detail structures…” were corrected as “…preserve the detailed structure…”

43. Line 310: the statements of “…except removes…” were corrected as “…except for it removes…”

44. Line 313: replace “denosing” -> “denoising”

45. Response to comment: (Line 320: Could the authors find a more descriptive subtitle heading which indicates the content of the subsection.)

Response: “Our New Work” were revised as “The Innovation of Our Work”

46. Line 326: the statements of “…by kurtosis…” were corrected as “…by the kurtosis…”

47. Line 326: the statements of “…is a four-order…” were corrected as “…is a fourth –order…”

48. Response to comment: Line 329: “…kurtosis is below….”. Could you here provide your exact stopping criterion. I would guess the the kurtosis is below some very small value.

Response: Line 329: the statements of “…kurtosis is zero” were corrected as “…kurtosis is below 0.001”

49. Line 338: replace “indexs” -> “indexes”

50. Line 342: the statements of “it shoule be…” were corrected as “it should be…”

51. Line 348: replace “denosing” -> “denoising”

52. Line 350: replace “denosing” -> “denoising”

53. Line 352: replace “denosing” -> “denoising”

54. Response to comment:( Line 359: It is not clear what “the vague effect is”. Could you elaborate on this?)

Response: OBNLM has the vague effect, one can see there exist gray value ‘drifts’ in the denoised results.

55. Line 363: the statements of “…object of interested to potential…” were corrected as  “…object of interest to potential…”

56. Line 373: the statements of “it shoule be…” were corrected as “it should be…”

57. Line 405: replace “denosing” -> “denoising”

58. Line 405: replace “figure 1” -> “figure 11”

We appreciate for your warm work earnestly, and hope that the correction will meet withapproval.

Once again, thank you very much for your comments and suggestions.

Yours sincerely,

Mei Gao

Name: Baosheng Kang
